# The Maize Clade A PP2C Phosphatases Play Critical Roles in Multiple Abiotic Stress Responses

**DOI:** 10.3390/ijms20143573

**Published:** 2019-07-22

**Authors:** Zhenghua He, Jinfeng Wu, Xiaopeng Sun, Mingqiu Dai

**Affiliations:** 1National Key Laboratory of Crop Genetic Improvement, Huazhong Agricultural University, Wuhan 430070, China; 2Hubei Key Laboratory of Food Crop Germplasm and Genetic Improvement, Food Crops Institute, Hubei Academy of Agricultural Sciences, Wuhan 430064, China

**Keywords:** *Zea mays*, phosphatase, *ZmPP2C-A*, abiotic stresses, natural variation

## Abstract

As the core components of abscisic acid (ABA) signal pathway, Clade A PP2C (PP2C-A) phosphatases in ABA-dependent stress responses have been well studied in *Arabidopsis*. However, the roles and natural variations of maize PP2C-A in stress responses remain largely unknown. In this study, we investigated the expression patterns of *ZmPP2C-As* treated with multiple stresses and generated transgenic *Arabidopsis* plants overexpressing most of the *ZmPP2C-A* genes. The results showed that the expression of most *ZmPP2C-As* were dramatically induced by multiple stresses (drought, salt, and ABA), indicating that these genes may have important roles in response to these stresses. Compared with wild-type plants, *ZmPP2C-A1*, *ZmPP2C-A2*, and *ZmPP2C-A6* overexpression plants had higher germination rates after ABA and NaCl treatments. *ZmPP2C-A2* and *ZmPP2C-A6* negatively regulated drought responses as the plants overexpressing these genes had lower survival rates, higher leaf water loss rates, and lower proline accumulation compared to wild type plants. The natural variations of *ZmPP2C-As* associated with drought tolerance were also analyzed and favorable alleles were detected. We widely studied the roles of *ZmPP2C-A* genes in stress responses and the natural variations detected in these genes have the potential to be used as molecular markers in genetic improvement of maize drought tolerance.

## 1. Introduction

Reversible protein phosphorylation is one of the most universal post-translational modifications, which is involved in many signal pathways. Protein kinases (PKs) and protein phosphatases (PPs) work antagonistically in this process. Both families were originally discovered in animal systems, and researchers in recent years have revealed the significant structure and function difference of the two gene families in plant [1]. In *Arabidopsis*, more than 1000 kinase genes have been annotated, while only 112 phosphatase gene have been identified [2]. Based on the different substrates of phosphatases, plant PPs are classified into serine/threonine (Ser/Thr) phosphatases and tyrosine phosphatases. Ser/Thr phosphatases are further categorized into two subfamilies, PPs consisting of PP1, PP2A, PP2B, PP4, PP5, PP6, and PP7, and PPs containing PP2C and other metal ion-dependent phosphatases [3]. According to the result of phylogenetic analysis, the PP2C family was further divided into 11 subclades (A-K) [4]. A series of studies on *PP2C-A* genes uncovered that *PP2C-As* participate in the core regulatory network in ABA responses as negative regulators [2,3,5,6].

Osmotic stress caused by drought leads to the accumulation of ABA in plant. ABA is the key regulator of plant drought resistance [7]. The core signaling pathway of ABA containing three family genes, pyrabactin resistance1/PYR1-like/regulatory components of ABA receptors (PYR1/PYL/RCAR) receptor, PP2C phosphatases, and sucrose non-fermenting-1 related protein kinase 2 (SnRK2) kinases. PP2Cs interact with SnRK2s to inhibit the activities of SnRK2s under normal conditions, while the accumulation of ABA caused by drought stress leads to the interaction between PYR/PYL/RCAR receptors and PP2C phosphatases, thus, SnRK2 kinases are released to phosphorylate downstream ABA-responsive elements-binding protein/ABA-responsive elements-binding factor (AREB/ABF) proteins [5,8,9,10]. Apart from ABA signaling pathways, mitogen-activated protein kinase (MAPK) signaling pathways are also important parts in response to abiotic stress, and crosstalk between them has been reposted. Dephosphorylation of MAPKs catalyzed by PP2Cs results in the inactivation of MAPKs [11]. In addition, autophagy plays roles in abiotic stress responses including oxidative stress, drought, and salt [12]. Two PP2Cs were reported to promote macroautophagy by dephosphorylating Atg1 complex [13]. Besides, tremendous transcription factors (like MYB, WRKY, NAC, and so on) play vital roles in regulating plant response to abiotic stress [14,15].

There are nine members in clade A of PP2C in *Arabidopsis*, namely ABI1, ABI2, HAB1, HAB2, AHG1, AHG3/AtPP2CA, HAI1, HAI2, and HAI3 [7]. ABI1 and ABI2 were firstly identified by screening ABA insensitive mutants [16]. HAB1 was also a member reported early. HAB1 and HAB2 are both homology to ABI1. The protein sequence of HAB1 is 55% and 54% identical, respectively, to those of ABI1 and ABI2. Nevertheless, expression of HAB1 is induced by ABA treatment [17]. Over-expression of HAB1 in *Arabidopsis* results in tolerance to ABA, both in seeds and vegetative tissues, while the mutant of HAB1 shows the contrary phenotype to ABA. Both results demonstrate the negative role of HAB1 in ABA signal transduction [18]. Among the ABA hypersensitive mutants, *ahg3* displays the strongest phenotype during germination, but no significant phenotypes were observed in adult plants, implying the different functional patterns of PP2C-A individuals [19]. Similar to *ahg3*, *ahg1* was hypersensitive to ABA, NaCl, KCl, mannitol, glucose, and sucrose in seedling stage, but showed no difference to wild-type at adult stage. Moreover, *ahg1 ahg3* double mutant showed stronger phenotypes to its single parental mutant, suggesting that AHG1 and AHG3 work together to regulate the ABA signal pathway [20]. All nine *PP2C-A* genes were upregulated after the treatment of ABA, among which three genes, named *HAI1*, *HAI2*, and *HAI3*, were strongly induced in the vegetative phase. No expression of *HAI1* was detected in seeds, and the interaction between HAI1 and SnRK2 was clarified by bimolecular fluorescence complementation (BiFC) experiment, revealing HAI1 function in the vegetative stage [21]. Compared to other members, mutants of *HAI* genes showed the most modest phenotype to ABA treatment. Single *hai* mutant had no difference to wild-type in ABA sensitivity, while the double and triple mutants were insensitive to ABA in germination stage, but sensitive to ABA in post-germination stage [22]. All these studies from *Arabidopsis* indicated pleiotropic roles of clade A PP2C phosphatases in plant development and stress responses. How clade A PP2C phosphatases in maize (*ZmPP2C-As*) regulate stress responses and agronomic traits remain largely known.

Few progresses have been made in maize *PP2C-A* research in recent years. *ZmPP2C-A10* was identified through a candidate gene association analysis and was proven to be associated with drought tolerance. Similar to the mechanism in *Arabidopsis*, ZmPP2C-A10 is involved in ABA signaling pathway by interacting with some ZmPYL and ZmSnRK2 [23]. In research about maize protein phosphatase gene family, 16 *ZmPP2C-A* gene were identified [24]. So far, a total of 13 *ZmPYLs* and four *ZmSnRK2s*, together with 16 *PP2C-As*, in maize were identified, and their patterns of gene expression, subcellular localization and interaction network were investigated. All interaction combinations between ZmPYLs and ZmPP2Cs, and ZmPP2Cs and ZmSnRKs were detected by yeast two-hybrid assays, resulting in the discovery of a comprehensive core ABA signaling network in maize [25].

Many natural variations in drought tolerance in crops were discovered through association analysis in recent years [26]. With a 82-bp miniature inverted-repeat transposable element (MITE) inserted in the promoter, the expression of *ZmNAC111* was repressed via RNA-directed DNA methylation and H3K9 dimethylation, contributing to the reduced drought tolerance in maize [27]. The expression of *ZmVPP1* is regulated by a 366-bp insertion in the promoter containing three MYB cis elements, resulting in the improved drought tolerance in maize inbred lines with tolerant allele [28]. In the 5′UTR region of *ZmPP2C-A10*, a deletion of endoplasmic reticulum stress response element (ERSE) related to endoplasmic reticulum stress-induced gene expression, resulted in more tolerance of maize plant to drought stress [23]. These studies imply the feasibility of association studies in the discovery of natural variations concerned with complex agronomic traits in maize.

The main purpose of this study was to investigate the function of *ZmPP2C-As* in response to multiple abiotic stresses. Expression patterns of *ZmPP2C-As* with multiple stress treatments had been detected. Overexpressing most of *ZmPP2C-As* in *Arabidopsis* led to the discovery of their broad roles in stress responses. In addition, candidate gene association analysis revealed the natural variations of *ZmPP2C-As* associated with drought tolerance and the development of other agronomic traits, which may be used as molecular makers in maize genetic improvement.

## 2. Results

### 2.1. Expression Analyses of ZmPP2C-A Genes under ABA Treatment

In a previous research, 13 putative maize genes were identified belong to *ZmPP2C* clade A gene family, namely *ZmPP2C-A1* to *ZmPP2C-A13*, and their expression levels in drought response were recovered [23]. In order to know how these genes respond to other abiotic stresses, we studied the gene expression patterns of *ZmPP2C-A* under ABA treatments excluding *ZmPP2C-A8* and *ZmPP2C-A9* due to their low expression levels in our experiments.

The results showed that most *ZmPP2C-A* genes were induced either in shoot or in root, or in both, by external ABA application except *ZmPP2C-A13* (Figure 1). After being treated with different concentrations of ABA, *ZmPP2C-A3*’s expression levels in shoot and root varied significantly in a concentration-dependent manner (Figure 1). Similar expression patterns were observed for *ZmPP2C-A1*(shoot), *ZmPP2C-A2* (shoot), *ZmPP2C-A5* (shoot), *ZmPP2C-A6* (root), *ZmPP2C-A7* (root), *ZmPP2C-A10* (root), and *ZmPP2C-A11*(shoot and root) (Figure 1), indicating significant roles of these genes in ABA response in shoot or root or both organs. These results revealed the different expression patterns among different genes and organs under ABA treatment and suggested the functional segmentation of *ZmPP2C-As* in ABA pathways (Figure 1).

### 2.2. Expression Patterns of ZmPP2C-A Genes in Response to Salt Treatment

We next investigated the expression patterns of *ZmPP2C-A* genes under sodium chloride (NaCl) treatment. Interestedly, we observed that the expression patterns of *ZmPP2C-A* genes varied significantly in response to salt treatments. As shown in Figure 2, *ZmPP2C-A1*, *ZmPP2C-A2*, and *ZmPP2C-A11* were induced by NaCl at lower concentrations (≤100 mM), while inhibited by NaCl at higher concentrations (≥150 mM). *ZmPP2C-A3* was only induced by NaCl at lower concentrations; nevertheless *ZmPP2C-A5* was suppressed by NaCl at the highest concentration. *ZmPP2C-A7* was induced by NaCl from 50 mM to 150 mM, with a peak occurring at 100 mM; *ZmPP2C-A4*, *ZmPP2C-A6*, and *ZmPP2C-A10* were more strongly induced by NaCl at higher concentrations; *ZmPP2C-A12* and *ZmPP2C-A13* showed opposite expression patterns in response to NaCl stresses, namely *ZmPP2C-A12* was inhibited while *ZmPP2C-A13* was induced by NaCl at higher concentrations. In addition, more than twenty-fold changes were observed in *ZmPP2C-A6* under treatment with 150 mM NaCl, which was outstanding among *ZmPP2C-A* genes (Figure 2). These expression patterns suggested different roles of *ZmPP2C-A* genes in regulation of maize salt stress responses.

### 2.3. Roles of ZmPP2C-A Genes in Regulation of ABA and Salt Stress Responses

To determine the biological roles of *ZmPP2C-A* genes in abiotic stress responses, we overexpressed *ZmPP2C-As* in *Arabidopsis*, and multiple stresses experiments were conducted on the transgenic plants. In *Arabidopsis* and other plants, ABA has been widely known as a negative regulator of seed germination [29,30]. Salt is a natural stress that inhibits seed germination [31]. To know how *ZmPP2C-As* regulate ABA response, seeds of Col and *ZmPP2C-A6* overexpression lines were sown on a Murashige and Skoog (MS) medium, supplemented with or without 1 μM ABA. After 48 h of cultivation, germination rates of both Col and *ZmPP2C-A6* transgenic seeds were almost 100% on MS plates without ABA (Figure 3a,b). Although germination rates of both materials were decreased in MS plates with 1 μM ABA, germination rates of Col seeds were reduced more as compared to those of *ZmPP2C-A6* overexpression lines (Figure 3a,c). These data suggested that *ZmPP2C-A6* is a positive regulator of ABA responses. On MS plates supplied with 150 mM sodium chloride, *ZmPP2C-A6* overexpression lines germinated faster than Col, although there were no different germination rates between Col and *ZmPP2C-A6* overexpression transgenic seeds on MS plates without salt (Figure 3a,d), suggesting a positive role of *ZmPP2C-A6* in plant salt tolerance. The similar germination tendencies were also observed for transgenic *Arabidopsis* plants overexpressing *ZmPP2C-A1* (Appendix A) or *ZmPP2C-A2* (Appendix A) under ABA and salt stresses. Taken together, these data implied that *ZmPP2C-A1*, *ZmPP2C-A2*, as well as *ZmPP2C-A6*, played positive roles in ABA and salt stress responses.

### 2.4. ZmPP2C-A Genes Act as Negative Regulators in Plant Drought Responses

Previously we reported that *ZmPP2C-A10* attenuated plant drought tolerance [23]. In order to know the roles of other *ZmPP2C-A* genes in regulation of plant drought tolerance, we treated the ZmPP2C-A transgenic *Arabidopsis* plants with drought stresses. Twenty-day-old seedlings of Col and *ZmPP2C-A6* overexpression lines were stressed with severe drought stress. After being re-watered, survival rates were counted. We observed that more than 70% seedlings of Col survived, while only ~20% seedlings of *ZmPP2C-A6* transgenic lines survived (Figure 4a–c), implying that *ZmPP2C-A6* negatively regulates drought resistance. Water loss of excised leaves from Col and *ZmPP2C-A6* transgenic lines were measured every 30 min. The results showed that *ZmPP2C-A6* transgenic lines lost water faster than Col (Figure 4d). In addition, the proline contents of the two materials were detected with or without drought stresses. Under well-watered conditions (without drought stress), no obvious difference was observed between the two materials (Figure 4e). But under severe drought stress, the levels of osmolyte proline in Col was significantly higher than those in *ZmPP2C-A6* overexpression lines (Figure 4e). Therefore, compared to Col, *ZmPP2C-A6* overexpression lines were more sensitive to drought stress, which resulted from faster water loss and less accumulation of osmolyte proline. We conducted similar drought stress and water loss assay to *ZmPP2C-A2* transgenic lines, and similar phenotypes were observed to these transgenic lines in response to drought stresses (Appendix A). All these data suggested that both *ZmPP2C-A2* and *ZmPP2C-A6* played negative roles in plant drought stress responses.

To further confirm the role of *ZmPP2C-A6* in drought stresses, the expression levels of several marker genes, including *RD22*, *ABI1*, *ABA1*, and *ABA3* which were known to respond to drought and ABA, were analyzed. Expression levels of these marker genes were measured in Col and *ZmPP2C-A6* transgenic lines, with or without drought stresses. We observed that the expression levels of these marker genes were significantly and dramatically inhibited in *ZmPP2C-A6* transgenic lines as compared to Col plants after drought stresses (Figure 5). These results suggest that the accumulation of ABA or ABA signaling were attenuated in *ZmPP2C-A6* overexpression lines, which resulted in the sensitivity of these lines to drought stresses.

### 2.5. Identification of Natural Variation in ZmPP2C-A Genes Associated with Stress Responses

Previously, an association mapping panel containing 527 maize inbred lines with various genetic backgrounds were collected [32], and 368 inbred lines of which were re-sequenced through RNA-sequencing [33]. In addition, 1.25 M high quality single nuclear polymorphisms (SNPs) were called out from this association panel by integrating reduced genome sequencing (GBS), high-density array technologies (600 K), and RNA-sequencing [34]. Survival rates of this association panel after salt and drought stresses were reported by us and other researchers recently [35,36]. In order to mine the natural variations of *ZmPP2C-A1*, *2*, *6* associated with drought and salt tolerance, SNPs located within *ZmPP2C-A* genes were collected, and association analyses were performed using these SNPs and the survival rates of 368 maize inbred lines stressed with drought or salt, respectively. Three algorithms, namely general linear model (GLM), GLM + Q, and mixed linear model (MLM), were used in these analyses. We found that there were no SNP located in these *ZmPP2C-A* genes that were detected with all three algorithms associated with survival rates after salt stress (Appendix A). Meanwhile, there were different amounts of SNPs within *ZmPP2C-A6* that were identified to be associated with survival rates after drought stresses with all three statistical models (Appendix A). The lead SNP (SNP3503) was located in the intron of *ZmPP2C-A6*, which has the A and C alleles (Figure 6a). According to this lead SNP, the 368 inbred lines could be divided into two groups, the A and C groups, and the survival rates between these two groups were significantly different in that plants with allele A had much higher survival rates than those with C alleles (Figure 6b). Based on the lead SNP, we analyzed the haplotypes (Hap) of *ZmPP2C-A6* in the association panel. A total of four Haps were detected, and Hap1 and Hap2 have obviously higher survival rates than those of Hap3 or Hap4 (Figure 6c).

## 3. Discussion

Plants have developed complicated regulatory networks to respond to abiotic stresses, and plant hormone ABA plays pivotal roles in these processes [7,37]. The core regulatory network of ABA consists of three members, PYR/PYL receptor proteins as receptors, Clade A PP2C phosphatases as negative regulators, and SNF1-related protein kinase 2 (SnRK2) as positive regulators [5,9,10]. Similar to the signal pathway in *Arabidopsis*, *ZmPP2C-A10* has been reported to take part in ABA signal transduction by interacting with ZmPYLs and ZmSnRK2s [23]. Furthermore, the research on gene expression, subcellular localization, and interaction network of ZmPYLs, ZmPP2Cs, and ZmSnRK2s resulted in the discovery of core ABA signaling networks in maize [17]. *ZmPP2C-A10* is a negative regulator in drought resistance [23], but the biological function of the rest of the members in the *ZmPP2C-A* gene family remains unclear. To solve this problem, we analyzed the expression patterns of *ZmPP2C-As* to multiple stresses, and overexpressed *ZmPP2C-As* in *Arabidopsis* to survey their roles in abiotic stress responses.

After ABA treatment, most *ZmPP2C-As* were induced either in different tissues or by different concentration of ABA except *ZmPP2C-A13* (Figure 1). *ZmPP2C-A12* was the only *ZmPP2C-A* that was not induced by any concentration of sodium chloride (Figure 2), suggesting its different role from other *ZmPP2C-As* in response to abiotic stresses. It will be important to study how *ZmPP2C-A12* functions in stress responses in future. In terms of fold changes in expression, the biggest change occurred in *ZmPP2C-A3* under ABA treatment (Figure 2), and in *ZmPP2C-A6* under salt treatment (Figure 2), indicating their vital roles in the corresponding stress responses. Differences in expression patterns among *ZmPP2C-As* suggested intricate regulatory mechanisms exist in these genes in response to abiotic stresses.

It has been reported that plant PP2C-As were involved in many abiotic stress response pathways [18,38,39]. To seek for the potential regulatory factors of *ZmPP2C-As*, we analyzed the 1.5 kb promoter region of all *ZmPP2C-As*. We found that all of these promoter regions have many kinds of potential transcription factor (TF) binding sites (Appendix A). While many TF binding sites like basic region-leucine zipper (bZIP), auxin response factor (ARF), and ethylene responsive factor (ERF) binding sites commonly exist in all *ZmPP2C-A* promoter regions, some of them only exist in part of them. Meanwhile, binding sites of many transcription factors involved in other pathways such as endoplasmic reticulum (ER) stress, auxin response, and ethylene response were also found in most of *ZmPP2C-A* promoters. These results suggested that as an upstream signal receptor element in ABA response pathway, *ZmPP2C-As* may be regulated by many other abiotic stress response pathways and further change the phosphorylation level of downstream targets. Our study indicates the important role of ZmPP2CAs in crosstalk between abiotic stress signaling.

The germination experiments showed that, compared to the Col, *ZmPP2C-A6* overexpression lines were insensitive to ABA and salt (Figure 3). The same phenotypes were observed in *ZmPP2C-A1* and *ZmPP2C-A2* (Appendix A), demonstrating the functional redundancy among these *ZmPP2C-A* genes. As for drought stress, *ZmPP2C-A6* transgenic lines were more sensitive than the Col, resulting from the faster water loss and less accumulation of proline (Figure 4). The other members of ZmPP2C-As including *ZmPP2C-A2* (Appendix A) and *ZmPP2C-A10* [23] seem to play the similar roles in drought resistance, which is opposite to the roles of *ZmPYLs*, as *ZmPYLs* act as positive regulators in drought resistance [40]. Mutant analyses have been conducted to all members of *Arabidopsis* PP2C-As. Most single mutants displayed hypersensitivity to ABA in germination stage, and mutants of *AHG* were sensitive to salt and osmotic stress during germination [19,20,22]. As for function in abiotic stress, proline levels in *hai* mutants were higher than that in Col, indicating the negative role of *HAI* in drought resistance [22]. Additionally, double mutants of *Arabidopsis* PP2C-As were generated, and showed reduced water loss and resistance to drought stress [41]. Except for mutant analysis, *HAI1* overexpressing lines were insensitive to ABA treatment, and were sensitive to drought treatment [18]. The results are consistent with the phenotype of *ZmPP2C-A1, ZmPP2C-A2, ZmPP2C-A6* overexpression plants in our excrement, revealing that function of these PP2C-As are conserved between *Arabidopsis* and maize.

Four marker genes were chosen for expression analysis in *ZmPP2C-A6* transgenic plants, namely *RD22, ABI1, ABA1, and ABA3*. As a member of the BNM2, USP, RD22 and polygalacturonase isozyme (BURP) protein family, *RD22* was reported to be induced by drought, salt stress, and endogenous ABA [42], and has been used as a marker gene for abiotic stress [43,44,45]. *ABI1* is the key component in ABA signaling pathway in *Arabidopsis* [2,3,6], while *ABA1* and *ABA3* both participate in ABA biosynthesis [46]. All four marker genes were down-regulated in *ZmPP2C-A6* transgenic plants compared with Col plants after drought treatment, implying that both ABA biosynthesis and signaling were inhibited in *ZmPP2C-A6* transgenic plants. Overexpression of *ZmPP2C-A6* may lead to the negative feedback regulation of ABA accumulation and signal transduction in *Arabidopsis*. Considering the ABA function in promoting drought stress tolerance, the expression levels of marker genes were consistent with the phenotypes that *ZmPP2C-A6* transgenic plants were sensitive to drought stress.

To investigate the natural variation associated with drought and salt stress, candidate gene association analyses were performed between SNP within *ZmPP2C-As* and survival rate after salt or drought stress (Appendix A). The SNP3503 located in intron of *ZmPP2C-A6* was found to associate with drought stress, but the mechanism of how this SNP regulates *ZmPP2C-A6* function needs further studies. Hap1 and Hap2 were the favorable haplotypes in this locus, which may be useful in drought-resistant maize breading programs. In addition, candidate gene association analyses were also conducted between *ZmPP2C-A* genes and 17 maize agronomic traits, the associations were observed in 11 members (except ZmPP2C-A12, ZmPP2C-A13) of the gene family (Appendix A). These findings suggest that *ZmPP2C-As* may participate in maize growth and development, and the potential use of these natural variations in maize genetic improvement.

## 4. Materials and Methods 

### 4.1. Plant Materials and Growth Conditions

Maize inbred line B73 was cultured and used for gene cloning and expression analysis. B73 seeds were sown and grown in a greenhouse under 16 h light/8 h dark photoperiod at 28 °C. 

The *Arabidopsis* ecotype Col-0 was used as the wild-type. The pRCS2(bar)-ZmPP2C-As plasmids were introduced into Agrobacterium tumefaciens strain GV3101 and then transformed into Arabidopsis Col-0 ecotype using the floral dip method [47]. Seeds of transformed *Arabidopsis* were selected using MS medium supplemented with 25 mg/L hygromycin. Homozygous lines of T4 generations were used for further analysis.

*Arabidopsis* seeds were surface-sterilized with 10% bleach and washed three to four times with sterile water. The seeds were pretreated in a refrigerator at 4 °C for 3 days, then sowed on MS medium and placed in a plant growth incubator under a 16 h light/8 h dark photoperiod at 22 °C. After growing in the incubator for a week, *Arabidopsis* seedlings were transferred into soil in a growth chamber with the same light periods and temperature.

### 4.2. ABA and NaCl Treatments

For expression analysis, B73 inbred lines were grown in aerated Hoagland’s nutrient solution. The nutrient solution was replaced every three days. Seedlings at the three-leaf stage were subjected to two different treatments: ABA (0, 1, 10, 50 μM) and NaCl (0, 50, 100, 150, 200 mM). After 3 h, all samples collected were immediately frozen in liquid nitrogen and stored in a refrigerator at −80 °C until RNA extraction.

For the germination rate assay, full and uniform *Arabidopsis* seeds of wild-type and independent transgenic *ZmPP2C-A*-overexpression lines were evenly spotted in MS medium supplemented with 1 μM ABA or 150 mM NaCl. The normal MS medium was used as a control.

### 4.3. Germination Rate Analyses

For germination test, plants of wild-type and different transgenic lines were grown under the same growth chamber, and *Arabidopsis* seeds were collected at the same time. Seeds were put on plates containing MS medium with ABA or NaCl solution. Seed germination was determined based on the appearance of an embryonic axis protrusion, as observed under a microscope, and the germination rate was counted from the 2nd day to the end of the 5th to determine the percentage of stress-tolerant germinated seeds.

### 4.4. Drought Tolerance, Proline, and Water Loss Assays

After transplanting the seedlings into the soil, the wild-type and homozygous transgenic plants continued to grow under normal watering conditions in the growth chamber for 20 days. Watering was then halted. Samples were taken for marker gene and proline assays after 8 days. When plants began to exhibit lethal effects of dehydration after a further 2 weeks, watering was then resumed and the plants were allowed to grow for a subsequent 3 days, and the survival rate was investigated.

Proline content was measured using the colorimetric determination based on proline’s reaction with ninhydrin [48]. To assay proline, 50 mg of rosette leaves or cauline leaves were shredded into 10 mL centrifuge tubes using scissors. Each sample was incubated with 5 mL of 3% sulfosalicylic acid solution and boiled for 10–30 min to obtain the extract solution. After cooling to room temperature, 2 mL of supernatant were pipetted into a new 10 mL centrifuge tube and mixed with 2 mL acetic acid and 2 mL ninhydrin at 100 °C for 30 min. After mixture had cooled down, the proline was extracted by 4 mL methylbenzene and assayed by UV-vis spectrophotometry (UV-1800 spectrophotometer, Shanghai, China) at 520 nm.

For the water loss assays, the *Arabidopsis* rosette leaves with normal growth of about 20 days were detached on the filter paper to lose water, and the nodes were weighed at different times to calculate the water loss rate of the leaves. 

### 4.5. Constructs

For *Arabidopsis* transformation, the *ZmPP2C-A* genes were cloned from cDNA of maize line B73 individually, then digested with *EcoR* I and *Xho* I. The digested product was ligated to the binary vector pJET1.2 vector, and inserted into the pSAT6 vector to produce pSAT6-ZmPP2C-A. The expression cassettes using 2X35S promoters to drive expression of the *ZmPP2C-A* cDNAs were released from the pSAT6 vectors by digestion with PI-Psp I and inserted into the pRCS2-Bar-OCS binary vector and then transformed into *Arabidopsis* ecotype Columbia-0 [49]. 

### 4.6. RNA Purification and Expression Analysis

Samples were collected after ABA and salt treatments. Total RNA was extracted using Trizol reagent (TransGen, Beijing, China) from more than three seedlings for each treatment. RNA was treated with DNase I (Thermo Scientific, Waltham, MA, USA) for purification, and single-stranded cDNA was then synthesized using M-MLV reverse transcriptase (Promega, Madison, WI, USA). The maize and *Arabidopsis* actin genes were used as the internal control.

### 4.7. Association Analysis 

Candidate-gene association mapping was carried out to identify the causal variants of *ZmPP2C-A* in a set of 368 diverse maize lines. All phenotypic data used in the study were obtained in a previous study. Among 1.25 million high-quality SNP markers with Minor Allele Frequency (MAF) less than 0.05, 149 SNPs were found in the regions of *ZmPP2C-A1, ZmPP2C-A2, and ZmPP2C-A6* genes. Association analysis was performed using a mixed model (MLM), considering population structure and relative kinship, in TASSEL 5.0 (Available online: https://tassel.bitbucket.io/) [50].

### 4.8. Statistical Analyses

All of the experiments in this study were repeated three times, and the values presented are mean ± SD. Asterisks above the columns in figures indicate the significance of *T*-test, * *p* < 0.05, ** *p* < 0.01. Statistical analyses were performed using Excel 2013 (Microsoft, Redmond, WA, USA). Figures were plotted by using Adobe Photoshop CC 2018 (Adobe Systems, San Jose, CA, USA).

## Figures and Tables

**Figure 1 ijms-20-03573-f001:**
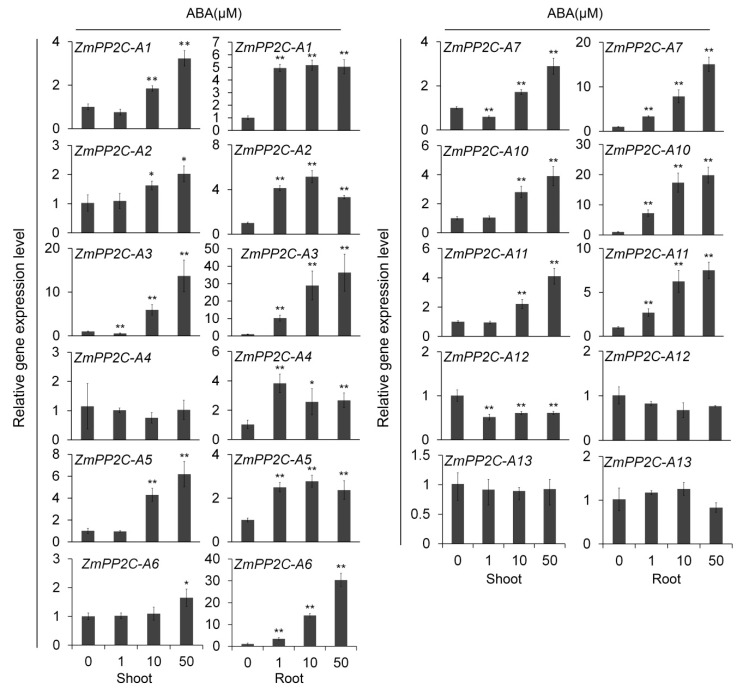
Relative expression of *ZmPP2C-A* genes to ABA treatment. The relative expression levels of *ZmPP2C-A* genes were analyzed by quantitative reverse transcription-PCR (qRT-PCR) in maize seedlings after ABA treatment. *ZmActin5* gene was used as an internal control. Seedlings at three-leaf stage were subjected to various concentrations of ABA (0, 1, 10, 50 μM) solution. Both leaf and root tissues were collected three hours after the treatments. Data represent the mean ± standard deviation (SD) of three replicates. Asterisks indicate the significance of *T*-test, * *p* < 0.05, ** *p* < 0.01.

**Figure 2 ijms-20-03573-f002:**
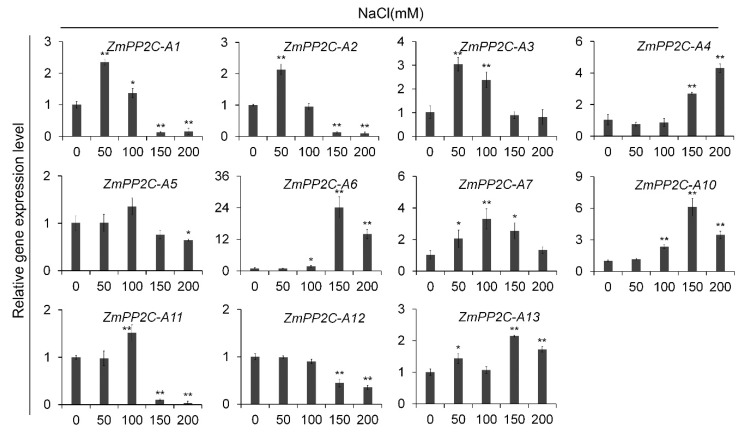
Relative expression of *ZmPP2C-A* genes to salt treatment. The relative expression levels of *ZmPP2C-A* genes were analyzed by qRT-PCR in maize seedlings after salt treatment. *ZmActin5* gene was used as an internal control. Seedlings at three-leaf stage were subjected with various concentrations of NaCl (0, 50, 100, 150, 200 mM) solution, and samples were harvested three hours after the treatments. Data represent the mean ± SD of three replicates. Asterisks indicate the significance of *T*-test, * *p* < 0.05, ** *p* < 0.01.

**Figure 3 ijms-20-03573-f003:**
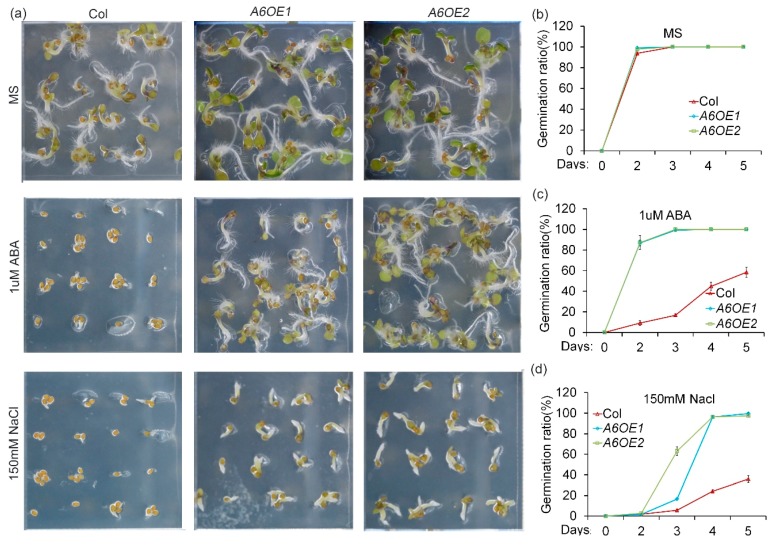
Overexpression of *ZmPP2C-A6* in *Arabidopsis* increased tolerance to ABA and salt stresses. Germination phenotypes (**a**) (Bar = 0.5 cm) and statistical analyses (**b**–**d**) of Col and two *ZmPP2C-A6* overexpressing lines (*A6OE1* and *A6OE2*) sown on MS medium (**b**) or MS medium supplemented with 1 μM ABA (**c**) or 150 mM sodium chloride (**d**). Data represent the mean ± SD of three replicates in (**b**–**d**).

**Figure 4 ijms-20-03573-f004:**
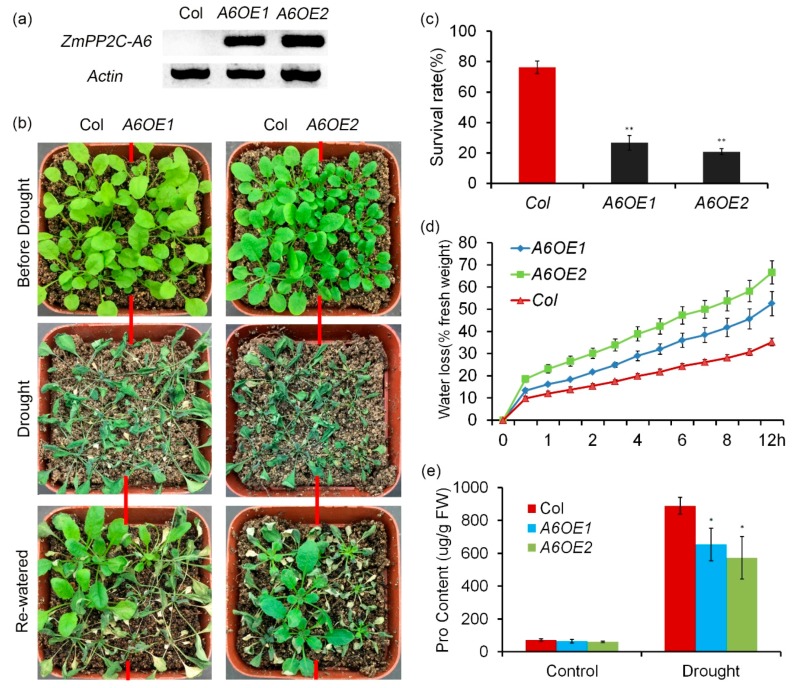
Overexpression of *ZmPP2C-A6* in *Arabidopsis* decreased tolerance to drought. (**a**) RT-PCR analysis of *ZmPP2C-A6* overexpression lines. (**b**) Drought tolerance assay of Col and two *ZmPP2C-A6* overexpression lines. Twenty-day-old plants were drought-stressed for ten days and then re-watered for three days. (**c**) Survival rates of Col and *ZmPP2C-A6* transgenic plants after drought stress. Asterisks indicate the significance of *T*-test, ** *p* < 0.01 (**d**) Water loss assay indicates that leaves from *ZmPP2C-A6* transgenic plants more easily lost water after detached. (**e**) Proline contents of well-watered and drought-stressed Col and *ZmPP2C-A6* transgenic plants. Asterisks indicate the significance of *T*-test, * *p* < 0.05 Data represent the mean ± SD of three replicates in (**c**–**e**).

**Figure 5 ijms-20-03573-f005:**
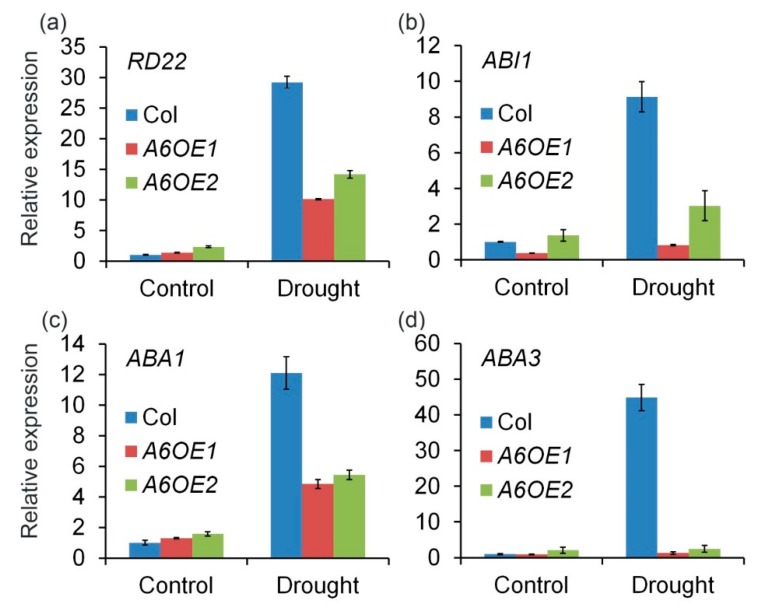
*ZmPP2C-A6* altered expression of stress-responsive genes in transgenic *Arabidopsis*. (**a**) The *RD22* were involved in responding to drought and ABA treatment; (**b**) The *ABI1* is the ABA signaling regulators gene which was also had a great effect on the ABA biosynthesis in the plants; (**c**,**d**) The *ABA1* and *ABA3* genes are the main function of ABA biosynthesis. Expression of stress-responsive genes in *ZmPP2C-A6* transgenic plants and Col. Overexpression of *ZmPP2C-A6* reduced the expression of drought responsible marker genes. The expression of all marker genes under normal and drought conditions was examined by qRT-PCR. Three-week-old plants were either watered or drought-stressed for 8 days.

**Figure 6 ijms-20-03573-f006:**
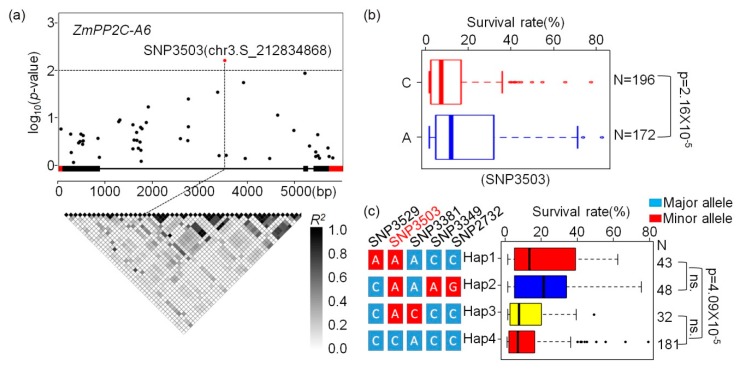
Association analysis of *ZmPP2C-A6* natural variations with drought tolerance. (**a**) *ZmPP2C-A6* association analysis and pairwise linkage disequilibrium (LD) among the SNP markers. The significantly associated SNPs are shown in red (*p* < 0.01). The gene structure is shown in the middle. Exons and introns are shown as filled boxes and dark lines, respectively. (**b**) Association of two alleles in SNP3503 with drought tolerance. (**c**) Association of haplotypes (Hap) in *ZmPP2C-A6* with drought tolerance. ns. means no significant.

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
