# Peer review of "The Maize Clade A PP2C Phosphatases Play Critical Roles in Multiple Abiotic Stress Responses"

_ijms, 2019, doi:10.3390/ijms20143573_

Round 1
Reviewer 1 Report
The paper aimed to investigate the expression patterns of ZmPP2C-As treated with multiple stresses and generated transgenic Arabidopsis plants overexpressing most of the ZmPP2C-A genes. The Authors have investigated an interesting topic and the theme has been properly described. I would like to congratulate authors for the quality of the article, the literature reported used to write their paper, and for the clear and appropriate structure. The manuscript is well written, presented and discussed, and understandable to a specialist readership. In general, the organization and the structure of the article are satisfactory and in agreement with the journal instructions for authors. The subject is adequate with the overall journal scope. The work shows a conscientious study in which a very exhaustive discussion of the literature available has been carried out. The introduction provides sufficient background, and the other sections include results clearly presented and analyzed exhaustively. Thus, I recommend the acceptance of the paper IJMS.
Author Response
--We sincerely thank the reviewers for all the valuable comments which make the manuscript more solid and significant. We have tried our best to conduct the essential experiments suggested by the reviewers. We hope that the revised manuscript is now suitable for publication at IJMS.
--We wish to thank this reviewer for his/her firm support.
Reviewer 2 Report
1. The
MS entitled "The maize clade A PP2C phosphatases play critical roles in
multiple stress responses" focused on the elucidation of the functions
of thirteen (eleven to be exact) maize phosphatases in different abiotic
stresses in Arabidopsis.
The title does not fit well with the overall contents and the findings of this MS. A quick look at the title would tell the readers that the study describes multiple stress responses (that means both abiotic and biotic). This renders the title a non-specific one. Hence the title of this MS should be rephrased to make a connection with the text.
2. The introduction section did not present a balanced review of the literature as expected. The first impression of the very first paragraph is really bad and brings reviewers attention more towards language rather the technical background knowledge on the subject. First, the authors introduced the gene family and suddenly authors started talking about the "catalytic activity" of the genes related to metal ions. PP2C phosphatases do play diverse roles such as they are also involved in autophagy, What relevance may be present between the macroautophagy and the role of PP2Cs observed by the authors?
The second thing which a reviewer/reader would be looking for in the review of the literature (i.e. the Introduction section) about the interplay of different genetic factors such as genes and transcription factors etc in the regulation of studied abiotic stresses. This seems missing from the introduction section of this manuscript and greatly limits the audience of this MS. I strongly argue that authors should include a review of literature discussing the pathways and genetic factors involved in abiotic stress responses at least in Arabidopsis. There are several good examples such as (transcription factors in abiotic stresses DOI: 10.1016/j.jplph.2018.04.007, PP2C in autophagy https://doi.org/10.1073/pnas.1817078116, etc.
There is little relevance between the Introduction and Discussion sections.
The English language of this MS needs very much attention. The results are good and must be supported however the authors were unable to present in a readable fashion.
The discussion section is very weak and must be improved in relevance to recent studies. It seems like authors are just summarizing the results rather than explaining their findings and showcasing the justifications. It is important to discuss the results/findings in relations to the orthologs (which are already characterized in Arabidopsis) and find a connection.
3. There are certain unanswered questions which should not be included in the discussion section and could be possibly moved to "conclusion" such as L249, L266, L272-273, and others.
4. P8, L248-249, What type of different role could be played by ZmPP2C-A12? Are there any structural domains or other elements in the sequence of the gene or the promoter region which rendering no induction in response to NaCl stress?
5. P8, L250-254. Authors must must must discuss in detail why the expression of individual gene differed in different abiotic stress and what regulatory elements might be involved in the differences in expression?
6. P9, L262-268. Where is the discussion? These are the results but authors need to discuss them properly with reasoning and justifications. I can't see any discussion. Same is the case with the last paragraph of the discussion section. Just authors stated one thing at the last sentence that "These findings suggest that ... may participate in maize growth and development, and the potential use of these natural variations in maize genetic improvement".
Author Response
1. This
is a good suggestion and we totally agree with this reviewer. We
changed the title to “The maize clade A PP2C phosphatases play critical
roles in multiple abiotic stress responses”
2. Thank
this reviewer for his/her critical reading and valuable suggestions for
the introduction and discussion sections. We revised the manuscript as
suggested.
3. We revised our manuscript carefully according to this reviewer’s valuable suggestions. Please see the revised manuscript.
4. We analyzed the 1.5kb promoter region of all ZmPP2C-As and found large amount of potential transcription factor binding sites (Figure
S 1 in our revised manuscript). We found that the binding sites of
ARR-B, CPP, EIL, GRF, LFY, NF-YB and RAV transcription factors are
absent in ZmPP2C-A12 promoter. The TFs, such as NF-YB and RAV,
have been reported widely involved in abiotic stress responses,
including salt stress response (Zhang et al. 2015; Duan et al. 2016).
Therefore, we deduce that these TFs could have a unique or combined
effect on the expression of ZmPP2C-A12.
5. We analyzed the cis-elements of 1.5kb promoter regions of each ZmPP2C-A gene,
and added this part of result in our revised discussion and
supplemental figures (revised Figure S 1). We deduced that the diversity
of ZmPP2C-As promoter region may finally cause the different role of ZmPP2C-As in salt stress response or other stress responses.
6. We
thank this reviewer for his/her valuable suggestions again, and added
proper discussion in the revised manuscript. Please see our revised
manuscript.
Reviewer 3 Report
The manuscript “The maize clade A PP2C phosphatases play critical roles in multiple stress responses” the authors analyse the role PP2C-As in abiotic stress responses in maize. Although this is an interesting topic, the manuscript has raised different concerns:
1. In figure 1, the authors claim that only ZmP2C-A12 expression is not induced after ABA treatment, but actually, there is no change in ZmP2C-A13 expression either. In addition, in some cases as ZmP2C-A12 and ZmP2C-A16 expression in shoots does not seem to be very pronounced. Statistics are needed to determine the significance of the differences in gene expression after the treatment.
2. Similarly, statistics are needed in figure 2, because for example I am not sure the differences in expression for ZmP2C-A15 are significant at all.
3. Why choosing A1, A2 and A6 for overexpression in Arabidopsis? ZmP2C-A1 and ZmP2C-A2 have very similar expression profiles after ABA and NaCl treatments, do it would have been more interesting choosing genes with different expression behaviour.
4. It is intriguing that lines overexpressing maize PP2C-As have a better germinating rate in the presence of ABA and NaCl, but they are more susceptible to drought. Both post-germinative growth and drought increase internal ABA levels, coordinating plant response, so it is shocking that the genes act as positive regulators of germination under ABA and osmotic stress, but as negative regulators of drought responses. This needs to be supported by additional experiments (i.e. gene expression after drought stress) and properly addressed in the discussion.
5. In figure 4 there is a faint band for ZmP2C-A6 in Col-0 and the actin band for A6OE1 is weaker than the one for Col-0 and A6OE2.
6. Also in figure 4, plants for A6OE2 are smaller than Col-0.
7. In general the English language can be improved, since some sentences are not very easy to understand.
Author Response
1. We are sorry for the misstatement. Actually only the expression of ZmPP2C-A13 is not induced after ABA treatment in both shoot and root. As there is no gene named ZmPP2C-A16 in our manuscript, we guess the gene mentioned by this reviewer could be ZmPP2C-A6. While ZmPP2C-A6 was slightly up-regulated under higher concentration of ABA (50 μM), ZmPP2C-A12 was down regulated under ABA treatment. To make our results clearer, we added statistics result in figure 1 and revised the description of the ZmPP2C-As’ expression patterns under ABA treatment. Please see details in our revised manuscript.
2. We performed statistical analysis for results in figure 2, please see the revised manuscript. As there is no gene named ZmPP2C-A15 in our manuscript, we think the gene reviewer has mentioned maybe ZmPP2C-A5, and ZmPP2C-A5 was only slightly down regulated at high concentration of NaCl (200 mM).
3. We agree with this reviewer that it would be more interesting choosing genes with different expression patterns in transgenic studies. In fact, we had constructed all ZmPP2C-As overexpression materials in Arabidopsis except A8 and A9 as we couldn’t clone it successfully, but only A1, A2 and A6 shown significant phenotype changes. We deduced that the genes without phenotype changes in transgenic studies might reflect functional divergence between different species.
4. In
fact, according to previous studies it is very important for plants to
sensing ABA signal when drought comes. Naturally, seed would stay in
dormancy stage as the concentration of ABA is high. But during
post-germinative growth, the internal ABA levels decrease dramatically.
For ZmPP2C-As overexpression lines, seed would not sense ABA
signal effectively enough and that would result in a better germinating
rate under ABA treatment compared with wild type. During drought stress,
higher ABA concentration leads to stomatal closure to protect plant from dehydration. As overexpression of ZmPP2C-As largely blocks ABA signaling, plant would loss more water from stomatal and be drought sensitive. Therefore, overexpression of ZmPP2C-As leads to be ABA insensitive in seeds and drought hypersensitive in seedlings.
5. We performed the expression experiment again and revise figure 4A. Please see the revised manuscript
6. It has been reported that PP2C-As in other plant species are involved in plant development. Overexpression of PP2C-As may cause developmental phenotypes, especially for the overexpression lines in which PP2CAs have higher expression levels. We have also observed this phenomenon in our previously study (Xiang et al. 2017). But even the leaf size of the transgenic plants was reduced, these leaves still lost water more quickly and accumulated less proline under drought stress as compared with wild types (Figure 4d).
7. We carefully polished our language as suggested. Please see our revised manuscript
Reference:
Zhang, T.; Zhang, D.; Liu, Y.; Luo, C.; Zhou, Y.; Zhang, L., Overexpression of a NF-YB3 transcription factor from Picea wilsonii confers tolerance to salinity and drought stress in transformed Arabidopsis thaliana. Plant physiology and biochemistry : PPB 2015, 94, 153-64.
Duan, Y. B.; Li, J.; Qin, R. Y.; Xu, R. F.; Li, H.; Yang, Y. C.; Ma, H.; Li, L.; Wei, P. C.; Yang, J. B., Identification of a regulatory element responsible for salt induction of rice OsRAV2 through ex situ and in situ promoter analysis. Plant molecular biology 2016, 90, (1-2), 49-62.
Xiang,
Y.; Sun, X.; Gao, S.; Qin, F.; Dai, M., Deletion of an Endoplasmic
Reticulum Stress Response Element in a ZmPP2C-A Gene Facilitates Drought
Tolerance of Maize Seedlings. Molecular plant 2017, 10, (3), 456-469.
Round 2
Reviewer 2 Report
The revision is satisfactory.
Author Response
We wish to thank this reviewer for his/her firm support.
Reviewer 3 Report
I thank the authors taking into consideration my comments. The revised manuscript still raises some concerns:
1. For ZmPP2C-A13 the authors mention in the text that it is the only gene without change in expression pattern, but later on they talk about different expression tendency in roots and shoots for ZmPP2C-A13.
2. The way the different expression patters after ABA or salt stress is explained in the text makes very difficult the understanding of the results for the reader. I.e. in lines 115-117 I am not sure what the authors want to point, if they are talking of a gene without significant different expression after ABA treatment.
3. Does the different expression behavior of the ZmPPC-2As correlate with the phylogenetic relationship? It would be very meaningful having some information about this matter.
4. When they authors talk about Col, do they mean Col-0?
5. In figure 3 it is not clear how many times has the experiment been done.
6. There are still some language errors, ad “seeding” instead of “seedling”.
7. In your answer you mentioned that only the Arabidopsis lines overexpressing A1, A2 and A6 show a clear phenotype, how can you explained the change in the expression of the rest of the genes after ABA of salt stress? They still should have a function related with these stresses.
Author Response
1. We thank the reviewer for his/her carefully check. We apologize for our ambiguous descriptive sentences. Actually we mean that ZmPP2C-A13 shows different expression pattern compared with other ZmPP2C-As as it shows no significant change under ABA treatment. Combined the result of expression level in root and shoot, we think ZmPP2C-As show different expression patterns among different genes and different tissues, and it may probably cause the function segmentation of ZmPP2C-As in ABA pathway. We also revised the corresponding sentences in our manuscript please see our revised manuscript for detail.
2. Like Q1, we revised the corresponding sentences in our manuscript and we wish to thank the reviewer again for his/her conscientious checking for our manuscript.
3. We think this is an interesting question. To ensure whether the expression patterns of ZmPP2C-As under salt or ABA treatment correlated with the phylogenetic relationship. We combined our result with previous works. But we found only part of ZmPP2C-As in adjacent branches show similar expression patterns. We think it maybe probably caused by the gene context change after the expansion of gene family. Actually this is a common phenomenon that the expression of the genes in same family shows different expression patterns.
4. Yes, we mean Col-0 and we have mentioned it in our materials and methods.
5. We have done 3 times for the experiment.
6. We thank for the reviewer for his/her check and we revised this typo in our manuscript.
7. We agree with the reviewer’s viewpoint that other ZmPP2C-As should also have a function related with ABA or salt stress as we found the expression most of ZmPP2C-As shows response to ABA or salt. But we only found only A1, A2 and A6 shows a clear phenotype, we think it was probably caused by the ectopic expression.